# Clustering via Self-Supervised Diffusion

Roy Uziel [1 2]  Irit Chelly [1 2]  Oren Freifeld [1 2 3]  Ari Pakman [4 3 2]

## Abstract

Diffusion models, widely recognized for their success in generative tasks, have not yet been applied to clustering. We introduce Clustering via Diffusion (CLUDI), a self-supervised framework that combines the generative power of diffusion models with pre-trained Vision Transformer features to achieve robust and accurate clustering. CLUDI is trained via a teacher–student paradigm: the teacher uses stochastic diffusion-based sampling to produce diverse cluster assignments, which the student refines into stable predictions. This stochasticity acts as a novel data augmentation strategy, enabling CLUDI to uncover intricate structures in high-dimensional data. Extensive evaluations on challenging datasets demonstrate that CLUDI achieves state-of-the-art performance in unsupervised classification, setting new benchmarks in clustering robustness and adaptability to complex data distributions.

## 1. Introduction

Clustering is a fundamental task in unsupervised learning, essential for uncovering meaningful groupings within data. These groupings play a vital role in diverse downstream applications, such as image segmentation (Mittal et al., 2022; Friebel et al., 2022), anomaly detection (Song et al., 2021a), and bioinformatics (Karim et al., 2021). Despite significant advancements, traditional methods face significant challenges, particularly in datasets with intricate structures and varying intra-class similarity, where such approaches often struggle to capture underlying patterns (Ben-David, 2018). To address these limitations, deep learning-based clustering

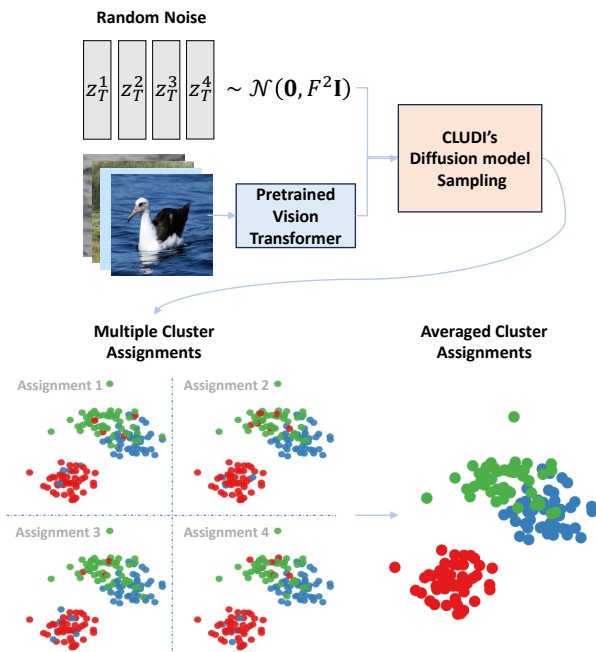

Figure 1: **Overview of the CLUDI Framework at Inference.** Images pass through a pre-trained Vision Transformer to obtain their feature representations. Multiple random vectors are sampled from a Gaussian distribution. A diffusion model, conditioned on the features, refines the random vectors into class assignment embeddings. Each refined embedding corresponds to a candidate cluster probability vector. By averaging multiple such assignments, the framework produces robust and accurate clustering predictions.

approaches have gained substantial attention for their ability to tackle complex and diverse data landscapes (Zhou et al., 2024; Ren et al., 2024; Wei et al., 2024).

However, deep learning-based clustering methods still face persistent challenges that limit their practical utility. A primary issue is *model collapse* (Amrani et al., 2022), where learned representations degenerate into trivial solutions. Another obstacle arises from underutilizing pre-trained features, resulting in suboptimal performance. Notably, Adaloglou et al. (2023) demonstrate that using pre-trained Vision Transformers (Caron et al., 2021) can surpass methods that attempt to learn both representations and cluster assignments simultaneously, thereby highlighting

---

[1]Department of Computer Science, Ben-Gurion University of the Negev, Beer Sheva, Israel [2]Data Science Research Center, Ben-Gurion University of the Negev, Beer Sheva, Israel [3]The School of Brain Sciences and Cognition, Ben-Gurion University of the Negev, Beer Sheva, Israel [4]Department of Industrial Engineering and Management, Ben-Gurion University of the Negev, Beer Sheva, Israel. Correspondence to: Roy Uziel <uzielr@post.bgu.ac.il>.

*Proceedings of the 42nd International Conference on Machine Learning*, Vancouver, Canada. PMLR 267, 2025. Copyright 2025 by the author(s).

the effectiveness of high-quality feature initialization. Self-supervised representations have consistently outperformed supervised ones in transfer tasks (Ericsson et al., 2021), highlighting the benefits of leveraging powerful pre-trained models. Meanwhile, many clustering frameworks rely on complex augmentation pipelines, such as neighbor mining (Van Gansbeke et al., 2020; Adaloglou et al., 2023), or employ multiple clustering heads trained in parallel, selecting the best-performing one at evaluation time (Adaloglou et al., 2023; Amrani et al., 2022). Although these strategies can improve accuracy, they introduce additional complexity and computational overhead during training.

To address these limitations, we introduce **Clustering via Diffusion (CLUDI)**, a self-supervised framework that leverages the generative strengths of diffusion models to produce robust cluster probabilities. As Figure 1 illustrates, CLUDI takes as input a pre-trained Vision Transformer feature vector $\mathbf{x} \in \mathbb{R}^n$ and refines an initial random assignment embedding vector through an iterative diffusion process. Conditioned on $\mathbf{x}$, this process evolves over multiple steps and culminates in a final **assignment embedding** $\mathbf{z}_0 \in \mathbb{R}^d$, which encodes the likelihood of each data point belonging to each of the $K$ clusters.

During inference, CLUDI generates multiple stochastic samples of $\mathbf{z}_0$ and aggregates the corresponding predictions. By averaging across diverse representations, this strategy mitigates uncertainty, uncovers subtle structures in high-dimensional feature spaces, and yields more stable and accurate cluster assignments, even in complex and diverse data scenarios.

Our model is trained using a self-supervised Siamese architecture comprising teacher and student branches. The teacher, implemented as a diffusion model, generates assignment embeddings $\mathbf{z}_0 \in \mathbb{R}^d$ and cluster probabilities $p(k|\mathbf{x})$ for $k \in \{1, 2, \ldots, K\}$, through stochastic sampling. These outputs, which capture diverse and complementary views of the data, serve as targets for the student.

The student adapts its predictions to match the teacher's outputs, enabling it to uncover meaningful and distinct clusters in the data. The training process optimizes two complementary objectives: an *asymmetric non-contrastive loss* (Grill et al., 2020; Chen & He, 2021; Caron et al., 2021) for the assignment embedding $\mathbf{z}_0$ and a *non-collapsing cross-entropy loss* for the cluster probabilities $p(k|\mathbf{x})$. To facilitate effective clustering and prevent trivial solutions, the cross-entropy loss incorporates a uniform prior over the data minibatch (Amrani et al., 2022), encouraging well-separated and diverse cluster assignments.

**Why use diffusion for clustering?**

Despite their success in generative tasks, diffusion models have not been explored for clustering until now. We ar-

gue, however, that their ability to model and sample from complex, high-dimensional distributions makes them particularly suited for clustering high-dimensional data, such as images. By iteratively refining noisy representations, diffusion models can uncover underlying structure and variability in the data, providing a natural mechanism for capturing meaningful cluster assignments. Their stochastic nature further enables robust and diverse clustering predictions.

**Our key contributions are as follows:**

- We introduce **Clustering via Diffusion (CLUDI)**, a novel framework that is the first to leverage diffusion models for the clustering task.

- We propose a **self-supervised training paradigm** based on a teacher-student architecture, where a diffusion model generates stochastic and informative cluster assignments to guide the student in learning meaningful cluster probabilities.

- We demonstrate the effectiveness of **CLUDI** through extensive evaluations on benchmark datasets, achieving state-of-the-art accuracy and robustness in unsupervised classification tasks.

## 2. Related Work

Deep learning-based models that learn to cluster are usually referred to as performing *deep clustering* or *unsupervised classification*. Cluster categories are learned during the training phase and remain fixed during inference. These methods have received much interest in the machine learning community. Comprehensive reviews on this vast field can be found in (Zhou et al., 2024; Ren et al., 2024; Wei et al., 2024).

**End-to-end training.** Most previous works learn simultaneously feature representations and cluster categories. An early work in this area is Deep Embedded Clustering (DEC) (Xie et al., 2016), which jointly optimizes feature learning through an autoencoder and assigns clusters using a Kullback-Leibler (KL) divergence-based loss. Although DEC can be effective in learning clusters, its performance is sensitive to initialization and prone to model collapse.

DeepCluster (Caron et al., 2018) takes a different approach by alternating between pseudo-label generation and feature representation refinement. The method iteratively updates cluster centroids and uses the centroids to assign pseudo-labels, which are then used to optimize the feature space. However, the computational overhead associated with generating and updating pseudo-labels can be significant, especially for large-scale datasets.

Invariant Information Clustering (IIC) (Ji et al., 2019) offers a complementary method by maximizing mutual in-

formation between different augmentations of the same data. Among probabilistic approaches, Variational Deep Embedding (VaDE) (Jiang et al., 2016) integrates variational autoencoders with Gaussian mixture models to learn probabilistic cluster assignments. Recent works leverage popular self-supervised approaches. Self-Classifier (Amrani et al., 2022) uses a Siamese network to simultaneously learn representation and cluster labels. To avoid degenerate solutions, it uses a variant of the cross-entropy loss which we adopt in our model.

**Training on pre-trained features.** The idea of decoupling feature learning from cluster learning has been advocated by SCAN (Van Gansbeke et al., 2020) and TwoStageUC (Han et al., 2020). These models, however, lack efficiency, as they learn features from scratch for every dataset.

The use of pre-trained features, particularly Vision Transformers (ViTs) (Dosovitskiy, 2020), was proposed by TSP (Zhou & Zhang, 2022) and TEMI (Ren et al., 2024). These approaches allow the model to focus on refining cluster assignments without relearning low-level features, significantly reducing the computational cost for large-scale datasets. Our CLUDI approach relies on a similar feature extraction backbone, based on the DINO model (Caron et al., 2021), but proposes a more refined clustering model and training setup that leads to superior results.

DeepDPM (Ronen et al., 2022), which can be used in either end-to-end fashion or with pretrained features, is a deep clustering method that infers $K$ and is inspired by a Dirichlet process mixture sampler (Chang & Fisher III, 2013; Dinari et al., 2019). However, unlike CLUDI, it makes a stringent assumption about the distribution of the features within each cluster.

**Amortized clustering.** In another approach, called *amortized clustering*, the model does not learn fixed cluster categories. Instead, it learns to organize full datasets into clusters discovered at test time. This approach allows for real-time adaptation of clusters as new data is introduced. It is thus a form of meta-learning (Hospedales et al., 2021), and research on this task is just beginning to unfold (Pakman et al., 2020; Jurewicz et al., 2023; Wang et al., 2024; Chelly et al., 2025).

## 3. Background

**Diffusion Models.** Denoising diffusion probabilistic models (DDPMs) are a recent pivotal shift in the landscape of generative modeling (Sohl-Dickstein et al., 2015; Ho et al., 2020), with considerable success across a range of applications, including image synthesis, audio generation, and molecular design (Song et al., 2021b; Dhariwal & Nichol, 2021; Nichol & Dhariwal, 2021; Rombach et al., 2022). In

DDPMs, an initial data sample denoted by $\mathbf{z}_0 \in \mathbb{R}^d$, is transformed into pure Gaussian noise, $\mathbf{z}_T \sim \mathcal{N}(\mathbf{0}, \mathbf{I}_d), \mathbf{z}_T \in \mathbb{R}^d$, through a sequence of incremental additions of Gaussian noise. This *forward process* is Markovian and defined by:

$$q(\mathbf{z}_t|\mathbf{z}_{t-1}) = \mathcal{N}(\mathbf{z}_t; \sqrt{1 - \beta_t}\mathbf{z}_{t-1}, \beta_t\mathbf{I}_d), \qquad (1)$$

where we introduced discrete time steps $t = 1 \ldots T$, we assume $T = 1000$, and $\beta_t$ is a predefined noise schedule. Note that the forward processes in Equation 1 allows closed-form sampling at any timestep $t$. Using the notation $\alpha_t := 1 - \beta_t$ and $\bar{\alpha}_t := \prod_{s=1}^{t} \alpha_s$, we have:

$$q(\mathbf{z}_t|\mathbf{z}_0) = \mathcal{N}(\mathbf{z}_t; \sqrt{\bar{\alpha}_t}\mathbf{z}_0, (1 - \bar{\alpha}_t)\mathbf{I}_d). \qquad (2)$$

DDPMs are a set of deep network models and sampling techniques which reverse this process: starting from a sample $\mathbf{z}_T \sim \mathcal{N}(\mathbf{0}, \mathbf{I}_d)$, one generates samples at earlier times until a sample $\mathbf{z}_0$ from the data distribution is obtained.

In the next section we present the particular model and sampling technique we use in CLUDI. For more details on diffusion models see recent overviews (Luo, 2022; Turner et al., 2024; Chan, 2024; Nakkiran et al., 2024). We use discrete time steps $t$, but continuous time formulations also exist (Song et al., 2021c). Note that our use of diffusions in continuous space to generate discrete data resembles their use to generate discrete language tokens (Dieleman et al., 2022; Gao et al., 2024; Gong et al., 2022; Li et al., 2022).

**Self-Supervision.** Self-supervised learning (SSL) has emerged as a powerful paradigm for learning from unlabeled data. The type of SSL that we employ uses a Siamese architecture (Chicco, 2021), where two views of the input produce different representations. The model is trained to ensure that these representations are informative and mutually predictable. A major challenge is avoiding model collapse, where the representations become mutually predictive by minimizing the information about the inputs.

Several mechanisms have been proposed in the SSL context to avoid collapse, such as contrastive losses (Chen et al., 2020; Jaiswal et al., 2020) or clustering constraints (Caron et al., 2018; 2021). In this work we adopt the teacher-student framework (Grill et al., 2020; Chen & He, 2021; Caron et al., 2021), in which the teacher model generates labels for the data which the student learns to predict. As we will detail in Section 5, we avoid collapse via stop-gradients, a predictor layer, and strong prior assumptions. We refer the reader to recent SSL surveys (Balestriero et al., 2023; Özbulak et al., 2023; Shwartz Ziv & LeCun, 2024; Gui et al., 2024) for thorough overviews.

## 4. Clustering via Diffusion

Clustering via Diffusion (CLUDI) is a latent variable model for classification of the form

$$p_\theta(k|\mathbf{x}) = \int d\mathbf{z}_0 \, p(k|\mathbf{z}_0)p_\theta(\mathbf{z}_0|\mathbf{x}) \simeq \frac{1}{B}\sum_{i=1}^{B} p(k|\mathbf{z}_0^i), \quad (3)$$

where the last term is a Monte Carlo approximation obtained from $B$ samples of $p_\theta(\mathbf{z}_0|\mathbf{x})$, a data-conditioned diffusion model that generates *assignment embeddings*. The classification head is a simple logit projection followed by a tempered softmax,

$$p(k|\mathbf{z}_0) = \frac{\exp(\frac{[\mathbf{L}\mathbf{z}_0]_k}{\tau})}{\sum_{j=1}^{K}\exp(\frac{[\mathbf{L}\mathbf{z}_0]_j}{\tau})} \quad (4)$$

$$\equiv \mathbf{u}_k. \quad (5)$$

where $\mathbf{L} \in \mathbb{R}^{K \times d}$. Figure 1 illustrates the averaging over $B = 4$ samples of $\mathbf{z}_0$, while Figure 2 shows classification accuracy as a function of $B$ for both clean and noise-augmented data.

Starting from an initial Gaussian sample

$$\mathbf{z}_T \sim \mathcal{N}(\mathbf{0}, F^2\mathbf{I}_d), \quad (6)$$

the diffusion model outputs $\mathbf{z}_0 \in \mathbb{R}^d$. For reverse sampling we adopt a stochastic version of the Denoising Diffusion Implicit Model (DDIM) (Song et al., 2021b), which allows sampling backwards at arbitrarily earlier times $s < t$ by running the backward dynamics

$$p_\theta(\mathbf{z}_s|\mathbf{z}_t) = \mathcal{N}(\mathbf{z}_s; \boldsymbol{\mu}_\theta(\mathbf{z}_t, \mathbf{x}, s, t), F^2\sigma_{s|t}^2\mathbf{I}_d), \quad (7)$$

for $s < t$, where

$$\boldsymbol{\mu}_\theta(\mathbf{z}_t, \mathbf{x}, s, t) = \sqrt{\alpha_s}\left(\frac{\mathbf{z}_t - \sqrt{1-\alpha_t}\,\epsilon_\theta^{(t)}(\mathbf{z}_t, \mathbf{x})}{\sqrt{\alpha_t}}\right)$$
$$+ \sqrt{1-\alpha_s-\sigma_s^2}\cdot\epsilon_\theta^{(t)}(\mathbf{z}_t, \mathbf{x}) \quad (8)$$

$$\sigma_{s|t} = \sqrt{\frac{1-\alpha_s}{1-\alpha_t}\left(1-\frac{\alpha_t}{\alpha_s}\right)}, \quad (9)$$

and we defined

$$\epsilon_\theta^{(t)}(\mathbf{z}_t, \mathbf{x}) = \frac{\mathbf{z}_t - \sqrt{\bar{\alpha}_t}\,\tilde{\mathbf{z}}_\theta(\mathbf{z}_t, \mathbf{x}, t)}{\sqrt{1-\alpha_t}}. \quad (10)$$

Here $\tilde{\mathbf{z}}_\theta(\mathbf{z}_t, \mathbf{x}, t) : \mathbb{R}^d \to \mathbb{R}^d$ is a network that predicts $\mathbf{z}_0$, and is trained by minimizing

$$\mathbb{E}_{\mathbf{z}_0,\mathbf{z}_t,t,\mathbf{x}}\left[w(t)\|\tilde{\mathbf{z}}_\theta(\mathbf{z}_t, \mathbf{x}, t) - \mathbf{z}_0\|^2\right]. \quad (11)$$

Here $w(t)$ are fixed weights. This loss is a weighted variational lower bound on the data log-likelihood. In modeling

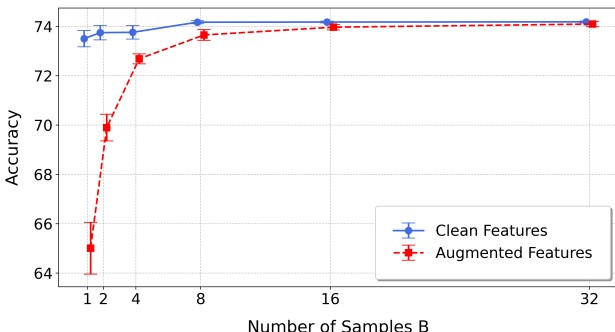

Figure 2: **Classification accuracy for clean and augmented inputs.** The model classification accuracy on augmented data (feature dropout plus Gaussian noise) becomes similar to that of clean data as the number of data samples grows. Results from ImageNet 100 validation data. Standard deviation based on 10 repetitions.

$\tilde{\mathbf{z}}_\theta$ we have followed (Salimans & Ho, 2022), but there exist other possibilities, such as modeling $\epsilon_\theta^{(t)}$, which represents the added noise in Equation 2.

We treat the noise scale $F^2$ in Eqs.(6)-(7) as an hyperparameter (Gao et al., 2024). Note that the above backward sampling requires choosing the values for the time steps in Equation 7, a freedom we exploit below in our training scheme.

**Noise schedule.** As in Li et al. (2022) we adopt the *sqrt* noise schedule with $\alpha_0 = \bar{\alpha}_0 = 1$ and

$$\bar{\alpha}_t = 1 - \sqrt{t/T + 0.0001}, \; t \geq 1. \quad (12)$$

As shown in Figure 3, this schedule accounts for the reduced sensitivity of the discrete labels to noise added near $t = 0$, leading to most of the noise in Equation 2 being introduced at lower $t$ values. As $t$ grows, the noise addition slows down, easing the learning process of the denoising network.

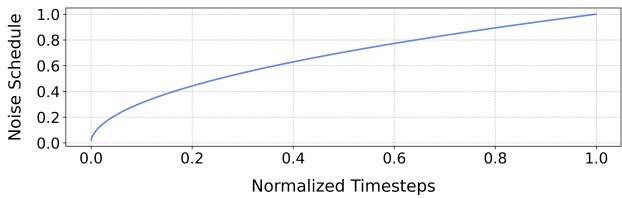

Figure 3: *sqrt* **noise scheduling.** The noise grows faster near $t = 0$, reflecting reduced sensitivity to noise at early timesteps, and grows gradually slower at later times.

## 5. Learning via Self-Distillation

CLUDI follows a self-supervised learning framework based on self-distillation, similar to BYOL (Grill et al., 2020), SimSIAM (Chen & He, 2021) and DINO (Caron et al., 2021).

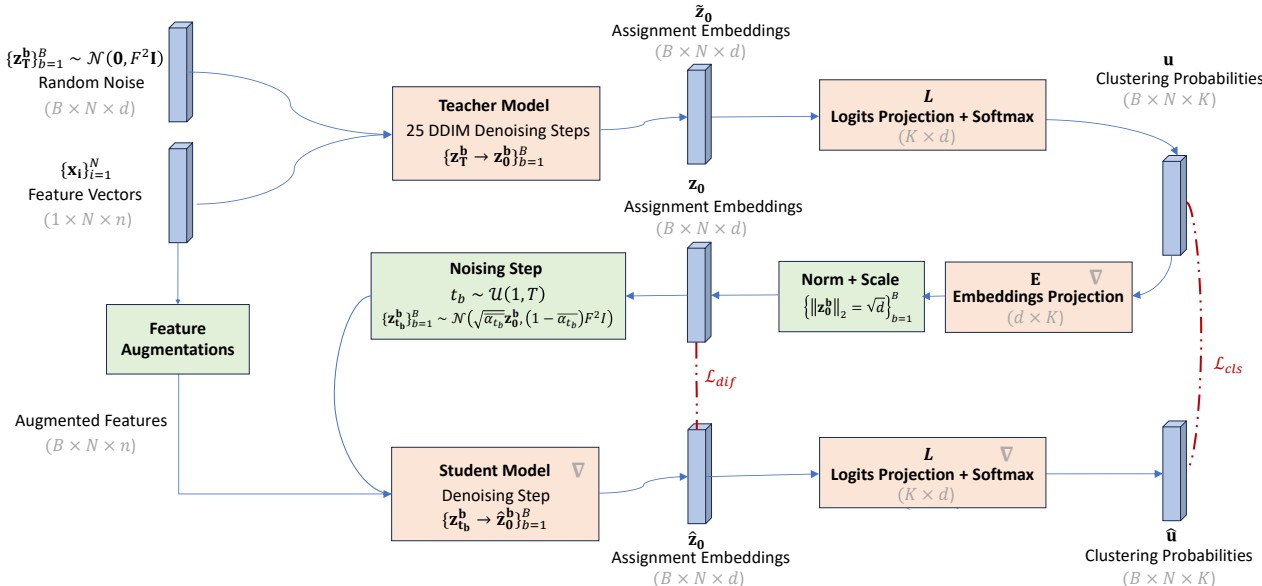

Figure 4: **Overview of CLUDI's training phase.** Given a set of image features $\mathbf{x}$, the teacher model generates denoised assignment embeddings $\tilde{\mathbf{z}}_0$ which are used to create two targets for the student: (i) clustering probabilities $\mathbf{u}$ and (ii) assignment embeddings $\mathbf{z}_0$, obtained from $\mathbf{u}$ via a predictor layer. The student network aims to predict both targets based on a version of $\mathbf{x}$ corrupted by feature dropout plus Gaussian noise. Components whose parameters are updated via gradient-based optimization are marked with the $\nabla$ symbol.

In this setup, two versions of the network process different representations of the same data $\mathbf{x}$, and one network predicts the output of the other. We adopt the SimSIAM approach, where the teacher and student networks share weights, unlike BYOL and DINO, where the teacher is updated via an exponential moving average of the student.

A key distinction of CLUDI is that, for each data point $\mathbf{x}$, the student has two different learning targets: the denoised assignment embedding $\mathbf{z}_0$ and the classification probabilities $u_k$ (Equation 5). Each target requires different strategies to avoid trivial solutions. For the embeddings $\mathbf{z}_0$, gradients are applied only to the student while keeping the teacher fixed, and a projection and normalization layer is added to structure the teacher's output into a useful learning target (Grill et al., 2020; Chen & He, 2021; Caron et al., 2021): These modifications prevent representational collapse and provide informative embeddings, though their precise effect remains an active area of research (Tian et al., 2021; Wang et al., 2021; Liu et al., 2022; Tao et al., 2022; Richemond et al., 2023; Halvagal et al., 2023).

To avoid collapse when learning the softmax cluster probabilities $\mathbf{u}_k$, we enforce a uniform prior over mini-batches that regularizes the cross-entropy loss (Amrani et al., 2022).

Figure 4 summarizes the sequence of teacher and student operations from the input data into the loss function. In the following section, we provide more details on our choices

for the teacher, the student, and the loss function.

## 5.1. Teacher model

The teacher generates cluster probabilities $\mathbf{u}$ and assignment embeddings, $\mathbf{z}_0$ which serve as training targets for the student. It receives $N$ input feature vectors $\mathbf{x}^i \in \mathbb{R}^n$, $i \in [1, N]$, and runs the denoising algorithm $B$ times for each of them, yielding $B \times N$ denoised embeddings $\tilde{\mathbf{z}}_0^{b,i} \in \mathbb{R}^d$. The time schedule of the teacher denoising is chosen to contain 25 equally-spaced timesteps from $t = T = 1000$ to $t = 0$, offering a balance between efficient denoising and capturing essential cluster characteristics. Acting on $\tilde{\mathbf{z}}_0^{b,i}$ with $\mathbf{L}$ and a softmax (see Equation 4-Equation 5), yields $B \times N$ probability targets $\mathbf{u}^{b,i} \in \mathbb{R}^K$ for the student.

Empirically, however, the learning is less effective when the student's targets directly the denoised embeddings $\tilde{\mathbf{z}}_0^{b,i}$. Instead, we map the probabilities $\mathbf{u}^{b,i}$ back to the embedding space by means of an embedding matrix $\mathbf{E} \in \mathbb{R}^{d \times K}$, and then normalize and scale the target embeddings as

$$\mathbf{z}_0^{b,i} = \sqrt{d}\frac{\mathbf{E}\mathbf{u}^{b,i}}{\|\mathbf{E}\mathbf{u}^{b,i}\|_2}. \qquad (13)$$

The role of $\mathbf{E}$ is akin to the predictor network in BYOL or SimSIAM, which assists in generating effective representations during training. Once the model is fully trained, however, $\mathbf{E}$ is no longer required. Given the above embed-

dings, the teacher class probabilities are

$$\mathbf{u}^{b,i} = \frac{\exp(\frac{[\mathbf{Lz}_0^{b,i}]_k}{\tau})}{\sum_{j=1}^{K} \exp(\frac{[\mathbf{Lz}_0^{b,i}]_j}{\tau})} . \quad (14)$$

### 5.2. Student model

The student network aims to predict both targets in Equation 13 and Equation 14. Since the input $\mathbf{x}$ consists of pretrained features, we use an abstract augmentation strategy. For each data feature vector $\mathbf{x}^i \in \mathbb{R}^D$, $i \in [1, N]$, we create $B$ augmented versions $\mathbf{x}^{b,i}$, $b \in [1, B]$, each obtained by first zeroing the components of $\mathbf{x}^i$ with probability 0.2, and then adding zero-mean Gaussian noise, with variance $\sigma^2$ sampled uniformly in $[0.1, 0.3]$. We present the student with noisy versions of the teacher's outputs in Equation 13,

$$\mathbf{z}_{t_b}^{b,i} \sim \mathcal{N}\left(\sqrt{\bar{\alpha}_{t_b}}\mathbf{z}_0^{b,i}, (1 - \bar{\alpha}_{t_b})F^2\mathbf{I_d}\right), \quad (15)$$

where $t_b$ is sampled uniformly from $[0, T]$. We denote the student predictions for the denoised assignment embedding and class probabilities as

$$\hat{\mathbf{z}}_0^{b,i} = \tilde{\mathbf{z}}_\theta(\mathbf{z}_{t_b}^{b,i}, \mathbf{x}^{b,i}, t_b), \quad (16)$$

$$\hat{\mathbf{u}}^{b,i} = \frac{\exp(\frac{[\mathbf{L}\hat{\mathbf{z}}_0^{b,i}]_k}{\tau})}{\sum_{j=1}^{K} \exp(\frac{[\mathbf{L}\hat{\mathbf{z}}_0^{b,i}]_j}{\tau})} . \quad (17)$$

### 5.3. Augmented views

Having presented the teacher and student models, we note that their different views of the data and assignment embeddings originate from (i) the stochastic nature of the teacher denoising, which starts with pure noise in Equation 6 and performs 25 sampling steps, (ii) the augmentation of the student feature vector (feature dropout + Gaussian noise) and (iii) the noise added in Equation 15 to the embedding that the student is required to denoise. We remark that unlike common augmentation strategies acting on raw images (cropping, color alterations, etc), our augmentations act directly on the data features or their assignment embeddings. In particular, the views generated by (i) and (iii) exploit the intrinsic randomness of the diffusion model.

### 5.4. Loss functions

**Embeddings.** We employ an MSE loss between the teacher denoised embeddings $\mathbf{z}_0^{b,i}$ from Equation 13 and the student-predicted embeddings $\hat{\mathbf{z}}_0^{b,i}$ from Equation 16:

$$\ell_{dif}(\mathbf{z}_0^{b,i}, \hat{\mathbf{z}}_0^{b,i}) = \|\mathbf{z}_0^{b,i} - \hat{\mathbf{z}}_0^{b,i}\|^2. \quad (18)$$

**Class probabilities.** We use here a more explicit notation for the teacher and student class probabilities, respectively,

as defined Equation 14 and Equation 17,

$$p(k|\mathbf{z}_0^{b,i}) = \mathbf{u}_k^{b,i}, \qquad p(k|\hat{\mathbf{z}}_0^{b,i}) = \hat{\mathbf{u}}_k^{b,i} . \quad (19)$$

Naively using a cross entropy loss,

$$\ell(\hat{\mathbf{z}}_0^{b,i}, \mathbf{z}_0^{b,i}) = -\sum_k p(k|\mathbf{z}_0^{b,i}) \log p(k|\hat{\mathbf{z}}_0^{b,i}), \quad (20)$$

quickly degenerates to a solution that puts all the probability on a single category. We regularize this loss using an idea from Amrani et al. (2022), which amounts to treating the indices $(b, i)$ as random variables with a uniform prior $p(\hat{\mathbf{z}}_0^{b,i}) = 1/(NB)$ and a distribution conditioned on $k$ given by a column softmax

$$p(\mathbf{z}_0^{b,i}|k) = \frac{\exp(\frac{[\mathbf{Lz}_0^{b,i}]_k}{\tau_{col}})}{\sum_{b',i'=1}^{B,N} \exp(\frac{[\mathbf{Lz}_0^{b',i'}]_k}{\tau_{col}})} , \quad (21)$$

with its own temperature $\tau_{col}$. Assuming also a uniform class prior $p(k) = 1/K$, Bayes rule and the law of total probability imply

$$p(k|\mathbf{z}_0^{b,i}) = \frac{p(\mathbf{z}_0^{b,i}|k)p(k)}{p(\mathbf{z}_0^{b,i})} = \frac{p(\mathbf{z}_0^{b,i}|k)}{\sum_{k=1}^{K} p(\mathbf{z}_0^{b,i}|k)} , \quad (22)$$

$$p(k|\hat{\mathbf{z}}_0^{b,i}) = \frac{p(k)p(k|\hat{\mathbf{z}}_0^{b,i})}{p(k)} = \frac{(NB/K)p(k|\hat{\mathbf{z}}_0^{b,i})}{\sum_{b',i'=1}^{B,N} p(k|\hat{\mathbf{z}}_0^{b',i'})} . \quad (23)$$

Inserting these expressions in Equation 20 leads to a loss which was shown in (Amrani et al., 2022) not to admit collapsed distributions as optimal solutions. In practice, we use a symmetric version of Equation 20,

$$\ell_{cls}(b, i) = \frac{1}{2}\left(\ell(\hat{\mathbf{z}}_0^{b,i}, \mathbf{z}_0^{b,i}) + \ell(\mathbf{z}_0^{b,i}, \hat{\mathbf{z}}_0^{b,i})\right). \quad (24)$$

**Weighting the minibatch elements.** Addressing the varying difficulty of the predictions of $\mathbf{z}_0^{b,i}, \mathbf{u}^{b,i}$ based on the sampled timestep $t_b$, we incorporate the Min-SNR-$\gamma$ (Hang et al., 2023) loss weighting strategy, which treats each timestep's denoising task as distinct and assigns weights based on their difficulty:

$$\text{SNR}_{t_b} = \frac{\bar{\alpha}_{t_b}}{1 - \bar{\alpha}_{t_b}}, \quad (25)$$

$$w_b = \frac{\max(\text{SNR}_{t_b}, \gamma)}{\text{SNR}_{t_b} + 1}, \quad (26)$$

where $\gamma$ is a predefined threshold set to 5, enhancing the stability and ensuring that no single noise level dominates during training. The overall loss function, which incorporates these weights, is structured as follows:

$$\mathcal{L} = \frac{1}{BN} \sum_{b,i=1}^{B,N} w_b \left(\ell_{dif}(\mathbf{z}_0^{b,i}, \hat{\mathbf{z}}_0^{b,i}) + \lambda\ell_{cls}(b, i)\right). \quad (27)$$

This formulation enables a nuanced control over the learning process, adapting to the challenges posed by different levels of noise.

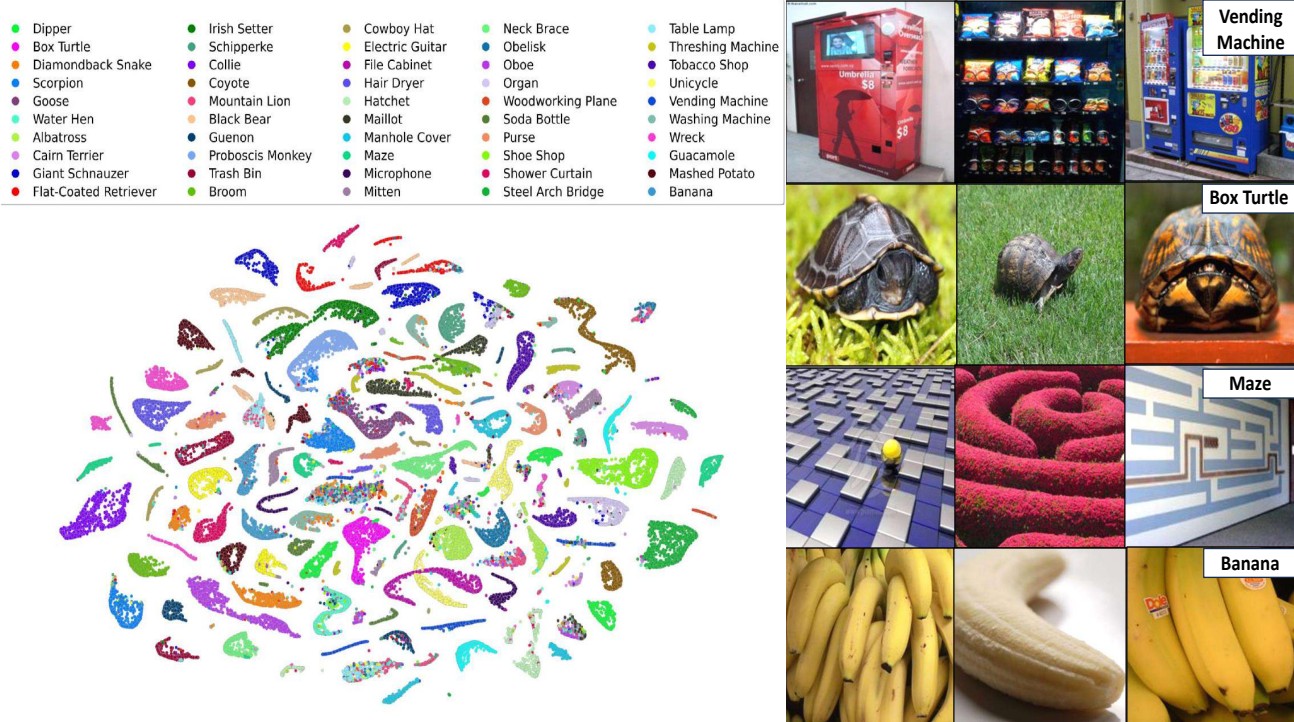

Figure 5: *Left:* t-SNE visualization of the assignment embedding space of ImageNet 50 demonstrating the model's ability to organize data points into well-separated clusters. *Right:* Examples of correctly classified images

## 6. Experiments

**Datasets.** We evaluate CLUDI on a comprehensive suite of benchmark datasets to rigorously assess its scalability, adaptability, and clustering performance. The datasets include subsets of ImageNet (Deng et al., 2009), Oxford-IIIT Pets (Parkhi et al., 2012) (with $K = 32$), Oxford 102 Flower (Nilsback & Zisserman, 2008), Caltech 101 (Fei-Fei et al., 2004), CIFAR-10 (Krizhevsky et al., 2009), and STL-10 (Coates et al., 2011). Each dataset introduces unique challenges: the ImageNet subsets cover broad and diverse categories, providing a robust test of scalability (see Table 2); Oxford-IIIT Pets and Oxford 102 Flower focus on fine-grained distinctions that assess CLUDI's precision; Caltech 101 evaluates generalization across diverse object types; and CIFAR-10 and STL-10 offer additional complex data that further validate CLUDI's ability to handle intricate clustering tasks (Table 1).

**Evaluation Metrics.** We use three popular metrics (Fahad et al., 2014): (i) *Normalized Mutual Information (NMI)* quantifies shared information between predicted and ground-truth clusters; (ii) *Clustering Accuracy (ACC)* measures the alignment of predictions with true labels; (iii) *Adjusted Rand Index (ARI)* adjusts for chance, providing a robust measure of similarity between predicted and true clusters.

**Experimental Setup.** For all experiments, we use features from DINO (Caron et al., 2021) based on Vision Transformers (ViT-S/16 and ViT-B/16) pre-trained on ImageNet. We present comparisons with four leading self-supervised clustering models: SCAN (Van Gansbeke et al., 2020), Propos (Huang et al., 2022), TSP (Zhou & Zhang, 2022) and TEMI (Adaloglou et al., 2023). We implement a variant of Self-Classifier (Amrani et al., 2022), denoted as *Self-Classifier\**, in which the feature extractor is frozen and only the classification heads are optimized. This setup ensures that CLUDI and Self-Classifier\* share identical feature representations, enabling a more direct and fair comparison.

All CLUDI results were obtained by running the embedding denoising for 100 equally spaced time steps. Figure 6 presents the ablation of the classification-loss weight $\lambda$, while Figure 7 illustrates the results of scanning different values of the embedding dimension $d$ in Equation 27. For additional implementation details and ablations, see our Appendix.

**Results.** CLUDI achieves state-of-the-art performance across all tested datasets, consistently outperforming previous approaches on clustering tasks of varying complexity. As detailed in Table 1 and Table 2, CLUDI significantly surpasses established baselines in NMI, ACC, and ARI, especially on ImageNet subsets, showcasing its robustness in both general and fine-grained clustering scenarios.

Table 1: **Clustering performances on smaller datasets.** The results for SCAN, Propos, TSP, and TEMI on CIFAR-10 and STL-10 are from the original papers, except for TEMI (ViT-S/16), which we trained using the official code. For the Oxford and Caltech datasets, we trained all models. The best result is shown in **bold**, the second best is underlined.

| Methods | NMI (%) | ACC (%) | ARI (%) |
|---|---|---|---|
| **CIFAR 10** | | | |
| SCAN (Resnet50) | 79.7 | 88.3 | 77.2 |
| Propos (Resnet18) | 88.6 | 94.3 | 88.4 |
| TSP (ViT-S/16) | 84.7 | 92.1 | 83.8 |
| TSP (ViT-B/16) | 88.0 | 94.0 | 87.5 |
| TEMI (ViT-S/16) | 85.4 | 92.7 | 84.8 |
| TEMI (ViT-B/16) | 88.6 | 94.5 | 88.5 |
| Self-Classifier* (ViT-S/16) | 83.8 | 91.2 | 82.1 |
| Self-Classifier* (ViT-B/16) | 84.2 | 89.0 | 80.3 |
| **Ours (ViT-S/16)** | 88.0 | 94.2 | 87.7 |
| **Ours (ViT-B/16)** | **89.6** | **95.3** | **89.8** |
| **STL 10** | | | |
| SCAN (Resnet50) | 69.8 | 80.9 | 64.6 |
| Propos (Resnet18) | 75.8 | 86.7 | 73.7 |
| TSP (ViT-S/16) | 94.1 | 97.0 | 93.8 |
| TSP (ViT-B/16) | 95.8 | 97.9 | 95.6 |
| TEMI (ViT-S/16) | 85.0 | 88.8 | 80.1 |
| TEMI (ViT-B/16) | 96.5 | 98.5 | 96.8 |
| Self-Classifier* (ViT-S/16) | 90.5 | 83.1 | 82.4 |
| Self-Classifier* (ViT-B/16) | 91.5 | 87.7 | 85.7 |
| **Ours (ViT-S/16)** | 95.7 | 98.2 | 96.1 |
| **Ours (ViT-B/16)** | **96.8** | **98.7** | **97.1** |
| **Oxford-IIIT Pets** | | | |
| TEMI (ViT-S/16) | 69.7 | 49.3 | 41.0 |
| TEMI (ViT-B/16) | 71.1 | 47.0 | 41.7 |
| Self-Classifier* (ViT-S/16) | 82.7 | 67.5 | 59.2 |
| Self-Classifier* (ViT-B/16) | 83.5 | 68.2 | 63.0 |
| **Ours (ViT-S/16)** | **87.3** | **74.1** | **71.6** |
| **Ours (ViT-B/16)** | 86.7 | 73.8 | 71.1 |
| **Oxford 102 Flower** | | | |
| TEMI (ViT-S/16) | 50.1 | 26.0 | 14.2 |
| TEMI (ViT-B/16) | 50.2 | 25.9 | 16.9 |
| Self-Classifier* (ViT-S/16) | 69.1 | 51.5 | 35.4 |
| Self-Classifier* (ViT-B/16) | 72.5 | 57.8 | 42.9 |
| **Ours (ViT-S/16)** | 76.1 | 62.2 | 52.6 |
| **Ours (ViT-B/16)** | **81.5** | **69.7** | **61.8** |
| **Caltech 101** | | | |
| TEMI (ViT-S/16) | 78.9 | 50.2 | 35.6 |
| TEMI (ViT-B/16) | 80.4 | 51.4 | 36.9 |
| Self-Classifier* (ViT-S/16) | 82.5 | 56.1 | 59.4 |
| Self-Classifier* (ViT-B/16) | 83.5 | 58.2 | 61.2 |
| **Ours (ViT-S/16)** | 86.5 | 66.7 | 65.7 |
| **Ours (ViT-B/16)** | **87.9** | **68.1** | **66.3** |

Table 2: **Clustering performances on ImageNet subsets.** The results for SCAN, Propos and TEMI are from the original papers. The best result is shown in **bold**, the second best is underlined.

| Methods | NMI (%) | ACC (%) | ARI (%) |
|---|---|---|---|
| **ImageNet 50** | | | |
| SCAN (Resnet50) | 82.2 | 76.8 | 66.1 |
| Propos (Resnet50) | 82.8 | - | 69.1 |
| TEMI (ViT-S/16) | 84.2 | 77.8 | 68.4 |
| TEMI (ViT-B/16) | 86.1 | 80.1 | 71.0 |
| Self-Classifier* (ViT-S/16) | 87.1 | 76.3 | 71.1 |
| Self-Classifier* (ViT-B/16) | 87.8 | 77.1 | 73.2 |
| **Ours (ViT-S/16)** | 90.1 | 81.3 | 75.8 |
| **Ours (ViT-B/16)** | **91.2** | **82.1** | **76.2** |
| **ImageNet 100** | | | |
| SCAN (Resnet50) | 80.8 | 68.9 | 57.6 |
| Propos (Resnet50) | 83.5 | - | 63.5 |
| TEMI (ViT-S/16) | 83.3 | 72.5 | 62.3 |
| TEMI (ViT-B/16) | 85.6 | 75.0 | 65.4 |
| Self-Classifier* (ViT-S/16) | 83.9 | 68.9 | 61.8 |
| Self-Classifier* (ViT-B/16) | 86.2 | 71.4 | 64.3 |
| **Ours (ViT-S/16)** | 86.5 | 74.3 | 67.5 |
| **Ours (ViT-B/16)** | **87.1** | **76.6** | **67.8** |
| **ImageNet 200** | | | |
| SCAN (Resnet50) | 77.2 | 58.1 | 47.0 |
| Propos (Resnet50) | 80.6 | - | 53.8 |
| TEMI (ViT-S/16) | 82.7 | 71.9 | 59.8 |
| TEMI (ViT-B/16) | 85.2 | 73.1 | 62.1 |
| Self-Classifier* (ViT-S/16) | 80.5 | 54.5 | 46.7 |
| Self-Classifier* (ViT-B/16) | 78.3 | 53.1 | 44.1 |
| **Ours (ViT-S/16)** | 85.9 | 73.2 | 60.9 |
| **Ours (ViT-B/16)** | **86.1** | **73.7** | **63.2** |

**Limitations.** The effectiveness of the CLUDI model is influenced by the choice of the diffusion parameter $F^2$ and the embedding dimensionality $d$, both of which play critical

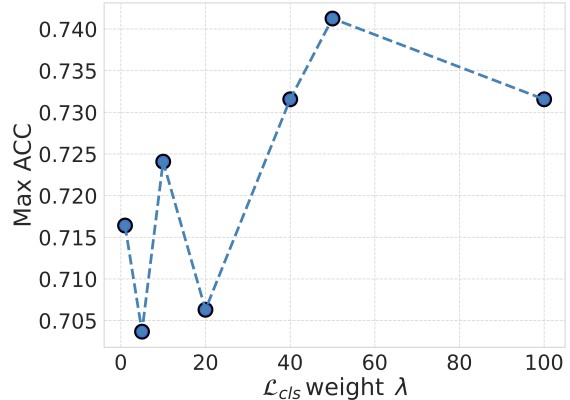

Figure 6: **Ablation Study on the $\mathcal{L}_{cls}$ weight $\lambda$.** Each point in the curves shows the maximum validation accuracy on ImageNet-100 achieved during training.

**Qualitative Analysis.** A t-SNE plot (Figure 5) of CLUDI's embeddings on ImageNet-50 demonstrates its ability to form well-separated clusters, reinforcing its quantitative metrics and illustrating its effectiveness in organizing complex data structures into distinct clusters. This visualization highlights CLUDI's capacity to capture subtle inter-class variations.

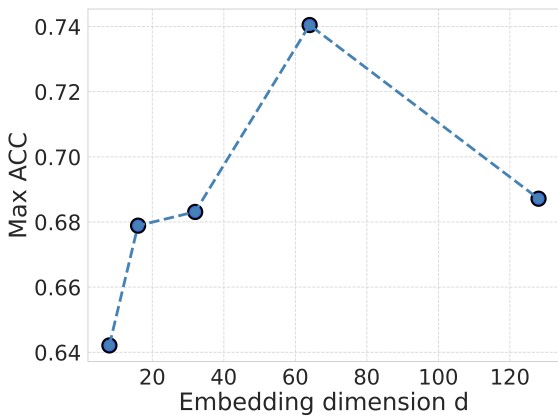

Figure 7: **Embedding dimension selection** $d$**.** Each point in the curves shows the maximum validation accuracy on ImageNet-100 achieved during training.

roles in determining clustering quality. While these parameters were tuned to optimize performance on the tested datasets, further refinement may be necessary for new data distributions or specific clustering tasks. Additionally, although CLUDI demonstrates strong capability in generating well-separated clusters, its performance can be impacted when scaling to a large number of clusters, as maintaining high-quality, distinct embeddings becomes increasingly complex with higher cluster counts.

## 7. Conclusion

In this work we introduced a novel use of diffusion models to generate clustering embeddings of pretrained data features. Our experimental results underscore CLUDI's advantages over both traditional and contemporary clustering techniques, validating its robustness, flexibility, and superior self-supervised clustering performance across diverse datasets and visual challenges. Future studies could build upon this work by investigating adaptive or data-driven hyperparameter selection techniques, as well as advanced clustering frameworks, such as hierarchical or multi-scale methods, potentially more scalable to large $K$ settings.

## Acknowledgments

This work was supported in part by the Lynn and William Frankel Center at BGU CS, by Israel Science Foundation Personal Grant #360/21, and by the Israeli Council for Higher Education (CHE) via the Data Science Research Center at BGU. A.P. was supported by the Israel Science Foundation (grant No. 1138/23). I.C. was also funded in part by the Kreitman School of Advanced Graduate Studies, by BGU's Hi-Tech Scholarship, and by the Israel's Ministry of Technology and Science Aloni Scholarship.

## Impact Statement

This paper presents work whose goal is to advance the field of Machine Learning. There are many potential societal consequences of our work, none which we feel must be specifically highlighted here.

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

In this Appendix we present the results of several hyperparameter ablations. All the curves shown correspond to performance metrics evaluated on the validation set of ImageNet 100. We also show some clustering examples on ImageNet 50.

## A. Hyperparameters

The model requires three hyperparameters: the embedding dimension $d$, the noise rescaling factor $F^2$ in Equation 6 and the coefficient $\lambda$ on the loss in Equation 27. A systematic scan on the validation set of ImageNet 100 yielded the optimal values $d = 64$, $F^2 = 25$ and $\lambda = 50$, which we adopted for all the datasets with $K \geq 100$. For datasets with fewer clusters (ImageNet 50, Oxford-IIIT Pets, STL 10, CIFAR 10) we used a smaller embedding $d = 32$.

## B. Ablation Study on $\mathcal{L}_{cls}$ weight $\lambda$

In Figure S1 we show the impact of varying the weight $\lambda$ associated with the classification loss $\mathcal{L}_{cls}$ on the overall clustering performance.

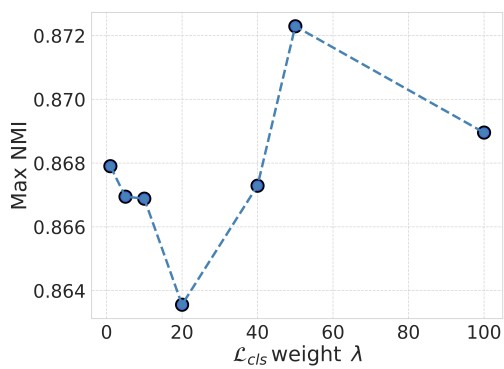

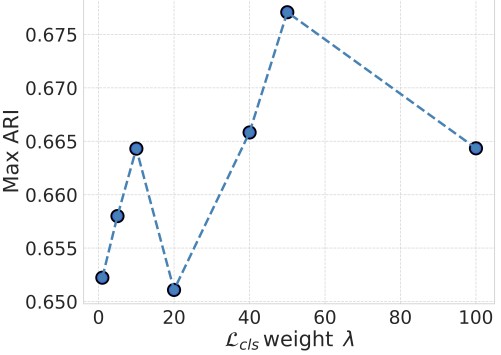

Figure S1: Clustering performance metrics (NMI, ACC, ARI) across $\lambda$ values for embedding dimension $d = 64$. Each image corresponds to a specific metric: (a) NMI, (b) ACC, and (c) ARI.

## C. Ablation Study on Rescaling Factor $F^2$ and Embedding Dimension $d$

In Figure S2 and Figure S3 we explore the effect on clustering performance of varying the rescaling factor $F^2$ and the embedding dimension $d$.

The noise rescaling factor $F^2$ plays a crucial role in our diffusion model by modulating the noise variance during the forward process. This parameter directly influences the model's ability to balance stability and exploration, which are essential for effective clustering. Insufficient noise (low $F^2$) results in trivial solutions where embeddings fail to explore the latent space adequately. Conversely, excessive noise (high $F^2$) destabilizes the learning process, disrupting inter-cluster separability.

Our results, consistent with those in (Gao et al., 2024) for text generation, confirm the need to find an optimal value (for us $F^2 = 25.0$) that strikes a balance between enabling the model to avoid degeneracy and preserving cluster coherence. In practical terms, our findings highlight that $F^2$ serves as a key hyperparameter for fine-tuning clustering models, with significant impacts on metrics such as NMI, ACC, and ARI.

## D. Clustering Visualization on ImageNet-50

In Figure S4 we provide a detailed visualization of the clustering results on the ImageNet-50 dataset. For ten different categories, we show images with highest and lowest confidence of belonging to each category. This illustrates the strength of our model and allows to visually explore the extent to which misclassifications are understandable.

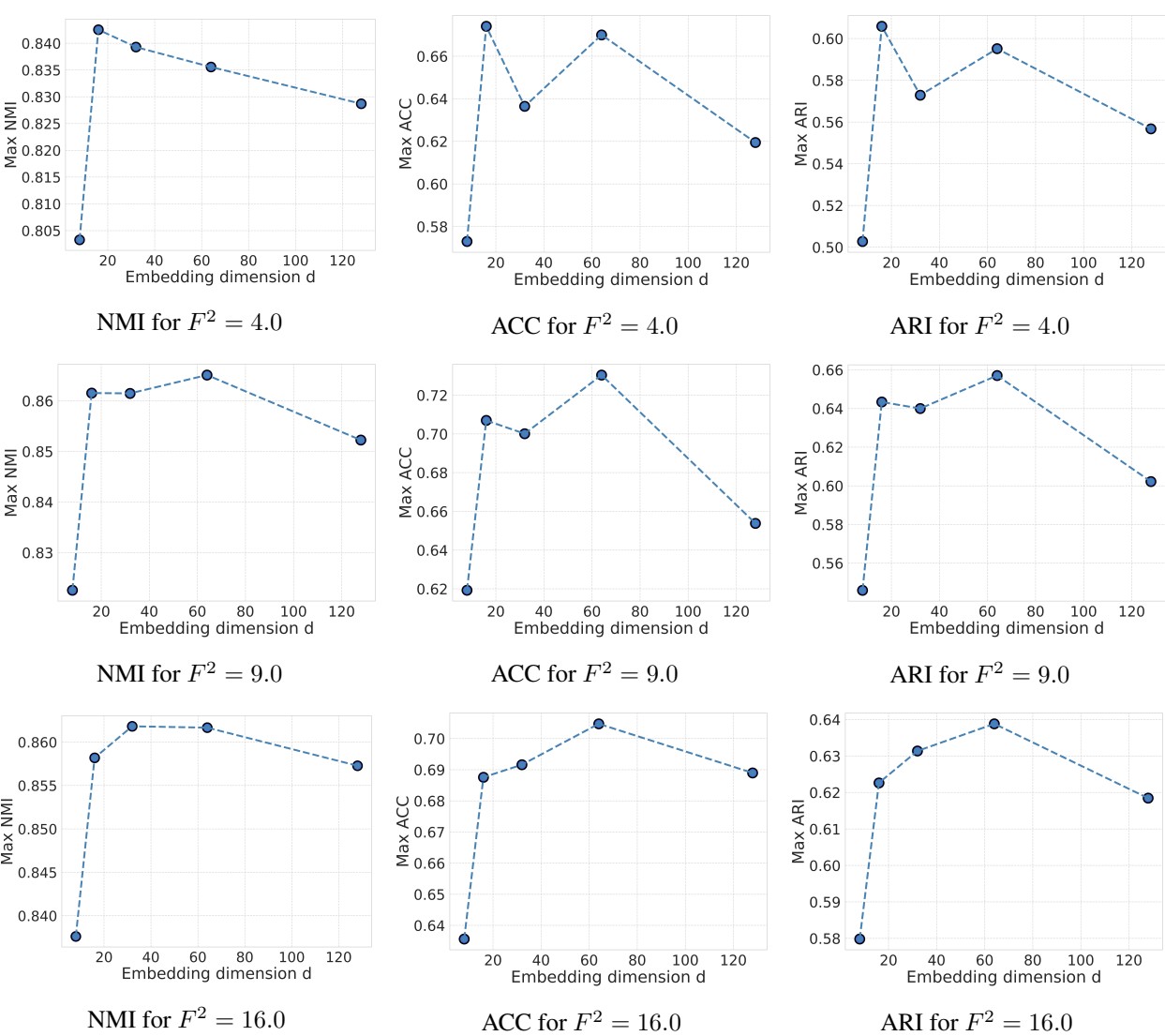

Figure S2: Clustering performance metrics for rescaling factors $F^2 = 4.0, 9.0, 16.0$ (top, middle, bottom, respectively) as a function of the embedding dimension $d$. Each column shows curves for a different clustering metric (NMI, ACC, and ARI).

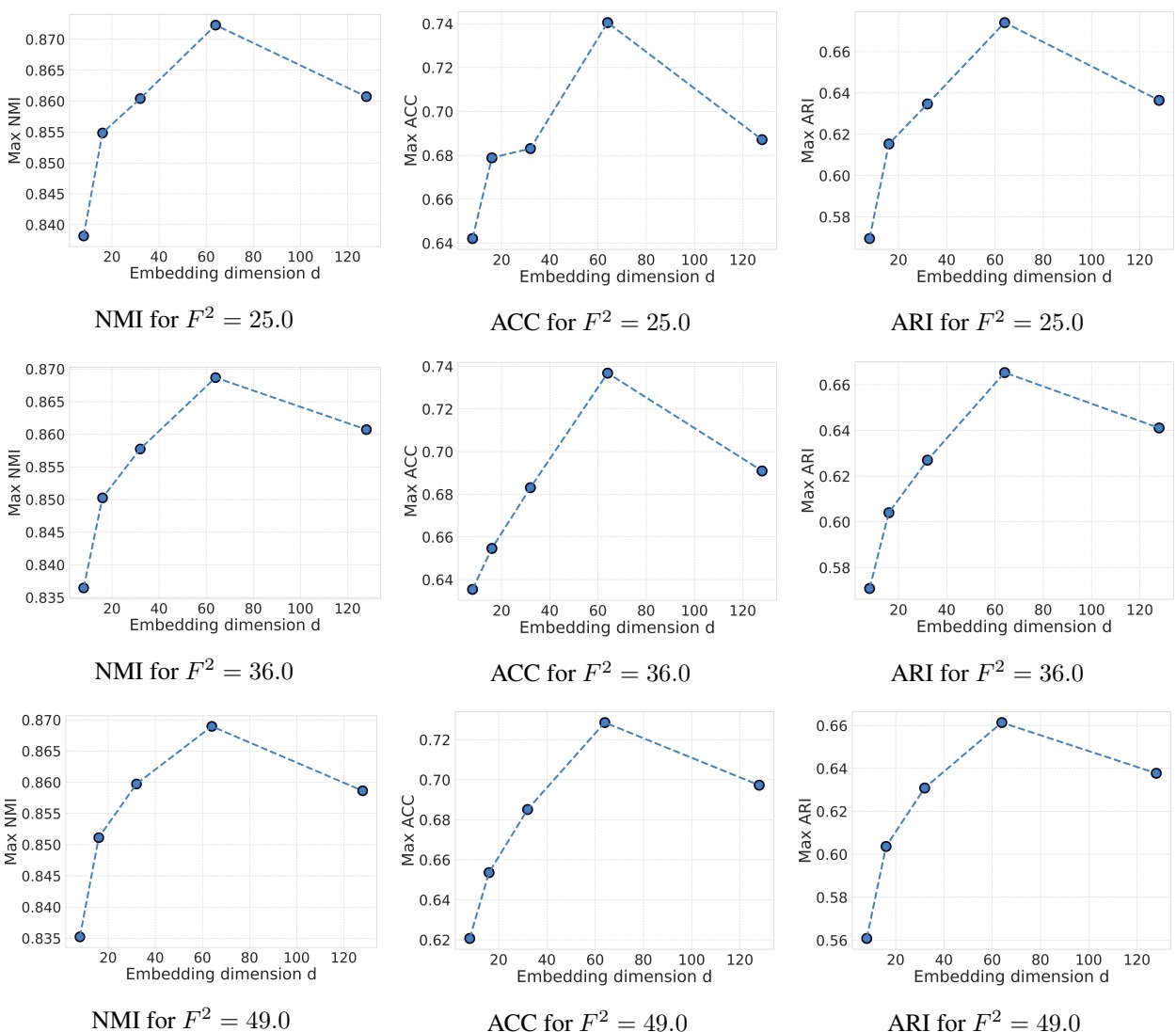

Figure S3: Clustering performance metrics for rescaling factors $F^2 = 25.0, 36.0, 49.0$ (top, middle, bottom, respectively) as a function of the embedding dimension $d$. Each column shows curves for a different clustering metric (NMI, ACC, and ARI).

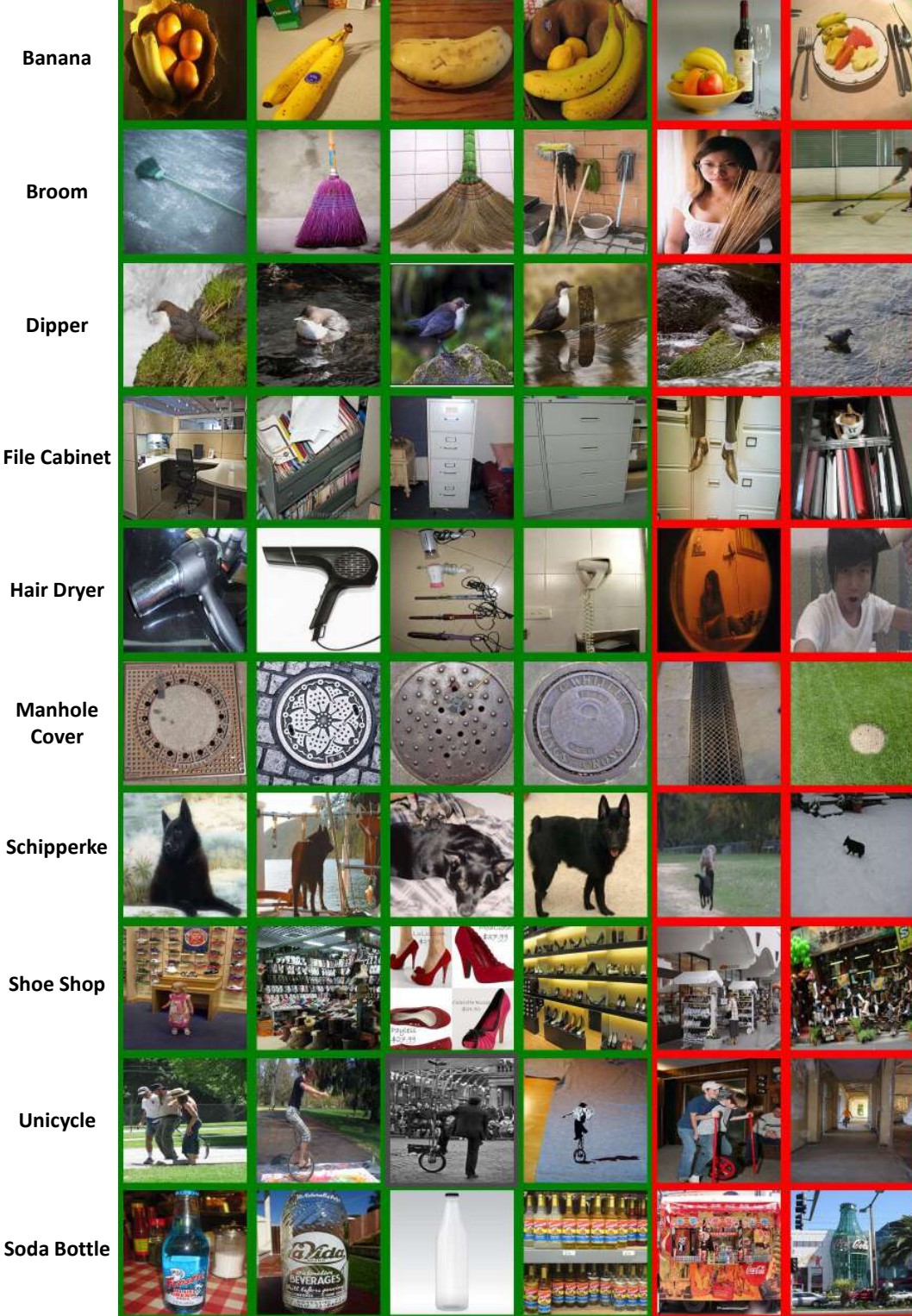

Figure S4: Visualization of supervised clustering results on the ImageNet 50 dataset. Each row corresponds to a single class, with the class name displayed on the left. For each class, the first four images represent the correctly classified samples with the highest confidence (outlined in green), while the last two images represent the incorrectly classified samples with the lowest confidence (outlined in red).

