# OpenReview forum: "Clustering via Self-Supervised Diffusion"
_ICML.cc/2025/Conference — ICML 2025 poster_

### Official Review · Reviewer_Azc9 · 2025-03-11

**Overall Recommendation:** 4

**Summary:**

This paper introduces Clustering via Diffusion (CLUDI), a self-supervised clustering framework that uses diffusion models on top of pre-trained Vision Transformer (ViT) features. The core idea is a teacher–student setup: a diffusion model (teacher) generates stochastic cluster assignment embeddings, while the student refines them into stable predictions. The paper claims this approach mitigates model collapse and underutilization of pre-trained features, yielding state-of-the-art clustering performance (NMI, ACC, ARI) across multiple datasets (ImageNet subsets, CIFAR-10, STL-10, Oxford Pets/Flowers, Caltech 101). The authors argue that diffusion’s ability to model high-dimensional distributions significantly enhances clustering robustness and accuracy.

## update after rebuttal
I have read the rebuttal to my review and found that the authors addressed the points within the expected scope. The response does not change my overall assessment, and I maintain my score of 4 (Accept).

**Claims And Evidence:**

- The authors claim that using a diffusion model as a teacher leads to better cluster assignment embeddings. They provide clear comparative results on multiple benchmarks (ImageNet-50/100/200, CIFAR-10, STL-10, Oxford datasets) showing consistent improvements in NMI, ACC, and ARI over existing methods (e.g., SCAN, TSP, TEMI, Self-Classifier*).
- They argue that combining a stochastic diffusion process with a uniform-prior cross-entropy loss avoids trivial solutions (collapse). Ablations and references to prior work (Amrani et al., 2022) support this claim.
- The authors state that using an embedding matrix E for learning, similar to BYOL and SimSiam, leads to better empirical performance compared to directly learning from the denoised embedding z₀ produced by the teacher model. However, the paper does not provide strong theoretical justification or detailed ablation experiments specifically analyzing the impact of E. It would have been beneficial to include such analyses to further support this design choice.

**Essential References Not Discussed:**

I do not see major omissions critical for contextualizing their key contributions. The authors cite a comprehensive set of core works in diffusion-based generative modeling and deep clustering.

**Experimental Designs Or Analyses:**

- Experiments are well-designed:
    - They compare CLUDI to multiple recent baselines under consistent settings (same backbone, same evaluation metrics).
    - They present ablations on the embedding dimension, diffusion noise scale (F²), and classification loss weight (λ).
    - They show t-SNE visualizations to illustrate cluster quality.
- The analyses appear valid, with no obvious methodological flaws or major confounding factors left unaddressed.

**Methods And Evaluation Criteria:**

- The authors employ standard clustering metrics: NMI, ACC, ARI. These are widely accepted for measuring how well predicted clusters align with ground truth labels.
- Benchmark Datasets are relevant and cover a spectrum of complexity: ImageNet subsets, small-scale (CIFAR-10, STL-10), and fine-grained tasks (Oxford Pets/Flowers).
- Freezing a pre-trained ViT and comparing classification heads ensures fairness in highlighting the contribution of the diffusion-based assignment component.

These methods and criteria make sense for evaluating clustering performance and appear appropriate for the task.

**Other Comments Or Suggestions:**

More details on training time or computational overhead vs. simpler clustering methods would be helpful.

**Other Strengths And Weaknesses:**

**Strengths**

- **Novelty**: Diffusion-based cluster assignments represent an innovative contribution.
- **Empirical Rigor**: SOTA results on diverse benchmarks, with thorough ablations.

**Weaknesses**

- **Limited Large-K Validation**: Results up to 200 clusters are strong, but the method’s performance on 500, 1000, or higher K remains underexplored.
- **Single Domain Focus**: Only tested on vision tasks, leaving question marks about text, audio, or other modalities.
- **Training Time & Computational Overhead**: The paper lacks a detailed analysis of how the diffusion-based approach compares to simpler clustering methods in terms of runtime and resource demands, especially at larger scales.

**Questions For Authors:**

No questions, as the paper's limitations are well-discussed, and there are no aspects that would change my overall evaluation.

**Relation To Broader Scientific Literature:**

This paper extends deep clustering literature (e.g., DEC, DeepCluster, VaDE, Self-Classifier) by incorporating a diffusion process for assignment embeddings. It also aligns with self-supervised learning frameworks (BYOL, SimSiam, DINO), adapting the teacher–student setup in a novel way.

**Theoretical Claims:**

- The paper relies mainly on established results in diffusion modeling (DDPM/DDIM) and non-collapsing self-supervised frameworks (BYOL, SimSiam).
- There is no novel theorem or proof that requires deep scrutiny. The authors do present equations for their backward sampling and training objectives, but these follow known formulations in diffusion literature.

---

> ### Author Rebuttal · Authors · 2025-03-30
>
> We thank the reviewer for the comments and questions.
>
>
> **Embedding-matrix ablations:** Our explorations on the learned projection matrix $\bf{E}$ yielded the following results for ImageNet 100:
> - Using the teacher output $\tilde{\bf{z}}_0$ directly as a target gives a drop in accuracy of around 4.7%.
> - Setting $\bf{E}=\bf{L}^T$ (removing the separate matrix) gives a drop in accuracy of around 3.2%.
> - Randomly-initializing and fixing $\bf{E}$ gives a 0.8% drop in accuracy.
>
> These results are consistent with findings in BYOL [1] and SimSiam [2]
>
> **Scalability to larger numbers of clusters:** We kindly refer the reviewer to our reply above to Reviewer 2.
>
> **Domain focus:** We focused on images because this is the domain where most comparable baselines exist for self-supervised classification. But our framework should similarly work on other modalities (e.g., language, audio), provided high-quality pretrained features are available.
>
>
>
>
> **Computational costs:** Bigger computational times, compared to  single forward pass models, are a known challenge of diffusion models. This is due to the sequential DDIM steps, which cannot be parallelized.
> Moreover, for *training* the memory demand is relatively high due to  large batch sizes (to maintain balanced assignments) and $B$ augmented views. These factors jointly increase both memory usage, and can be traded for gradient accumulation, increasing further the training time.
> For *inference*, we again do sequential DDIM denoising for each image, and sampling many embeddings scales linearly with the number $B$ of draws. This can be parallelized, but total cost still exceeds a single forward pass.
>
>
> [1] Grill et.al., Bootstrap your own latent - a new approach to self-supervised learning, NeurIPS 2020.
>
> [2] Xinlei Chen and Kaiming He, Exploring Simple Siamese Representation Learning, CVPR 2021.

---

### Official Review · Reviewer_6A3N · 2025-03-12

**Overall Recommendation:** 4

**Summary:**

This paper introduces a novel self-supervised image clustering framework incorporating the ideas of diffusion models to achieve accurate and robust clustering. The framework is designed in a teacher-student paradigm to train a teacher model to produce diverse cluster assignments and a student model for stable predictions. The effectiveness of the proposed framework has been validated on extensive benchmark datasets.

## update after rebuttal

This paper's ideas are interesting and novel to me. My questions have also been answered properly by the authors. I'd still recommend an acceptance.

**Claims And Evidence:**

One of the key advantages claimed for clustering with diffusion is its robustness. However, I failed to find any experimental evidence supporting this claim.

**Essential References Not Discussed:**

Yes

**Experimental Designs Or Analyses:**

I went through both the main experiments and ablation studies provided in the supplementary. Most of them are reasonable to me, with just a few concerns/questions:
1. How important is the Min-SNR-$\gamma$? I failed to find the ablation study of this.
2. Does the Self-Classifier perform better with trainable features? If so, how does CLUDI compare to Self-Classifier if both are with trainable features?
3. It is interesting to see that the ViT-S variant of CLUDI performs even better than the ViT-B variant on Oxford-IIIT Pets. Is there anything special about this dataset?

**Methods And Evaluation Criteria:**

Yes, the proposed method is interesting and technically solid to me. The evaluation criteria are standard in the field.

**Other Comments Or Suggestions:**

See comments above.

**Other Strengths And Weaknesses:**

The ideas of using diffusion models for image clustering is interesting and novel to me, the paper is generally well-structured and easy to follow. My main concerns and questions remain in the experiment parts as detailed above.

**Questions For Authors:**

See question in “Experimental Designs or Analysis”

**Relation To Broader Scientific Literature:**

The key contribution of this paper is to leverage the ideas of diffusion models for image clustering for stronger robustness and performance. This is novel to me and holds the potential to benefit also unsupervised representation learning that serves as the foundation for a lot of CV tasks by providing high-quality visual features.

**Theoretical Claims:**

N/A

---

> ### Author Rebuttal · Authors · 2025-03-30
>
> Thanks to the reviewer for the comments and questions.
>
> **Robustness:**
> The experimental evidence for the robustness of our approach is presented in Figure 2, which shows that when we corrupt the ViT inputs (via feature dropout + Gaussian noise), the degradation in classification accuracy is completely overcome by using big enough $B$ latent samples from the diffusion model.
>
> **Importance of the Min-SNR-$\\gamma$ weighting:** The table below shows an ablation study on the accuracy of ImageNet 100 (validation), with and without  Min-SNR-$\gamma$ weighting, as a function of the training epoch.
>
>
> | Training epoch | ACC (with Min-SNR-$\\gamma$) | ACC (without Min-SNR-$\\gamma$) |
> |---------------|--------------------------|-----------------------------|
> | 0             | 0.0100                   | 0.0100                      |
> | 1             | 0.6066                   | 0.5842                      |
> | 2             | 0.6860                   | 0.6648                      |
> | 3             | 0.7164                   | 0.6820                      |
> | 4             | 0.7132                   | 0.7036                      |
> | 5             | 0.7260                   | 0.6916                      |
> | 6             | 0.7196                   | 0.7002                      |
> | 7             | 0.7344                   | 0.7128                      |
> | 8             | 0.7320                   | 0.7110                      |
> | 9             | 0.7356                   | 0.7136                      |
> | 10            | 0.7330                   | 0.7146                      |
> | 11            | 0.7440                   | 0.7172                      |
>
>
> As is clear from these results, Min-SNR-$\\gamma$ speeds up convergence and yields higher final accuracy. These results are consistent with those in the work that proposed this weighting strategy [1], and will be included in the Supplementary Material.
>
>
>
> **Training features:** We fine-tuned Self-Classifier [2] on ImageNet 100 with a DINO backbone (so that augmentations are now on raw images rather than features), and obtained less than 1% improvement in accuracy relative to the frozen backbone. This is consistent with the observations in TEMI [3] that fine-tuning the ViT often yields marginal gains on large datasets. On the other hand, modifying CLUDI to fine-tune the entire backbone would require additional memory (since we maintain multiple augmented views and diffusion states), so for large-scale data it can become quite demanding.
>
>
> **Pets dataset:** This dataset is special because of its small size.
> It has about 200 images per class and small intraclass variance (compare with more than 1000 images per class for ImageNet and Flowers, with higher intraclass variance). Thus ViT-B is arguably overfitting the data here, as evidenced by ViT-S superior performance.
>
>
>
>
> [1] Hang et.al., Efficient Diffusion Training via Min-SNR Weighting Strategy, ICCV 2023.
>
> [2] Amrani et. al., Self-supervised classification network, ECCV 2022.
>
> [3] Adaloglou et. al., Exploring the limits of deep image clustering
> using pretrained models, BMVC 2023.

---

### Official Review · Reviewer_Qpdh · 2025-03-12

**Overall Recommendation:** 3

**Summary:**

This paper proposes Clustering via Diffusion (CLUDI), a method using diffusion models to cluster unlabeled image data. The authors take pre-trained ViT features as input, then learn a diffusion-based generative process that refines random noise into “assignment embeddings.” A classification head maps these embeddings to cluster-probability vectors. During inference, the model samples multiple such embeddings per image and averages their predicted cluster probabilities, which is shown to yield robust performance. The empirical results on CIFAR-10, STL-10, several ImageNet subsets (50, 100, 200 classes), etc., show that CLUDI outperforms a range of SOTA clustering baselines.

**Claims And Evidence:**

The claims made in the submission are supported by clear and convincing evidence.

**Essential References Not Discussed:**

the authors have a good coverage of references.

**Experimental Designs Or Analyses:**

The experimental design in this paper is well-structured.
This paper uses various image sets, including small CIFAR-10, moderate STL-10, medium/large ImageNet subsets and fine-grained sets. Then uses public pretrained ViT weights and only learns the diffusion-based assignment and classification heads. Also, the standard metrics, NMI, ACC, ARI, are computed and compared with previous approaches, including SCAN, TEMI, TSP, etc. Further, the authors also examine key hyperparameters, i.e., embedding dimension, noise scale, and weighting λ($\lambda$) of the classification loss.

**Methods And Evaluation Criteria:**

The proposed method is well-aligned with clustering in high-dimensional feature spaces by using pretrained ViT features from DINO. This paper employs standard unsupervised classification metrics for evaluation: NMI, Accuracy (ACC) (after best label alignment), ARI. The datasets used are well-established in image clustering, including CIFAR-10, STL-10, subsets of ImageNet, fine-grained sets like Oxford Pets/Flowers, etc., and the authors give ablation studies in different embedding dimensions, weight the classification loss, and rescaling factor.

**Other Comments Or Suggestions:**

See weakness above.

**Other Strengths And Weaknesses:**

Strengths:
This paper is well organized, and the proposed method is described clearly. As far as I know, this paper is the first application of diffusion-based generative modeling to produce embeddings for clustering.

Their experiments achieve top performance on multiple established benchmarks, CIFAR-10, STL-10, ImageNet subsets, etc.
Weakness:
The authors mention choosing 100 time steps in the reverse diffusion at inference. But it could be clearer on how the tradeoff changes with less step counts, like 25 steps, 50 steps.
The proposed method was found to be sensitive to hyperparameters.  With different size of datasets, careful tuning of the noise scale, embedding dimension, and classification loss weight λ($\lambda$) is required. This may complicate deployment on new datasets without a validation set.

**Questions For Authors:**

How does the approach scale beyond 200 clusters or to larger image sets like full ImageNet (1000 classes)? Is there an exponential blow-up in computation from multiple diffusion samples?
How sensitive is the model to choosing noise scale or weight λ\lambda when no labeled validation set is available?
Do you have any results for fewer inference steps?

**Relation To Broader Scientific Literature:**

This paper extends the well-known class of diffusion models (DDPMs, DDIMs) to a new, previously underexplored domain: unsupervised clustering. The proposed method employs a teacher–student setup in the style of BYOL, SimSIAM, and DINO, but rather than relying on a fully deterministic teacher network, it leverages a diffusion sampler to generate cluster assignments. Further, it aligns with methods such as DeepCluster, SCAN, and Self-Classifier, but crucially replaces fixed pseudo-labels & re-clustering and purely end-to-end latent clustering with diffusion-based assignment embeddings. Like TSP and TEMI, the proposed approach makes use of pretrained ViT features to avoid retraining low-level image representations.

**Theoretical Claims:**

There are no theoretical claims in this paper

---

> ### Author Rebuttal · Authors · 2025-03-30
>
> Thanks to the reviewer for the comments and questions.
>
>
> **Number of diffusion steps at inference:**
> Our experiments show that increasing the number of DDIM diffusion steps at inference reduces the variance of the accuracy across independent runs of the diffusion model. Moreover, for small latent samples $B$, the mean accuracy increases with more diffusion steps. To illustrate this, we present below mean accuracy and standard deviation (based on 10 independent runs) for the ImageNet 100 validation set, for different values of samples $B$ and DDIM diffusion steps.
>
>
> | Number of samples B | DDIM steps |  Mean accuracy    |   Std   |
> |---------|------------|----------|---------|
> | 1       | 5          | 0.7390   | 0.0031  | 0.7348  | 0.7420  |
> | 1       | 25         | 0.7403   | 0.0024  | 0.7385  | 0.7443  |
> | 1       | 50         | 0.7407   | 0.0018  | 0.7398  | 0.7451  |
> | 1       | 100        | 0.7409   | 0.0015  | 0.7400  | 0.7452  |
> |---------------------------------------
> | 2       | 5          | 0.7394   | 0.0028  | 0.7357  | 0.7425  |
> | 2       | 25         | 0.7406   | 0.0019  | 0.7386  | 0.7446  |
> | 2       | 50         | 0.7410   | 0.0015  | 0.7402  | 0.7453  |
> | 2       | 100        | 0.7411   | 0.0014  | 0.7403  | 0.7454  |
> |---------------------------------------
> | 4       | 5          | 0.7396   | 0.0026  | 0.7355  | 0.7427  |
> | 4       | 25         | 0.7407   | 0.0018  | 0.7388  | 0.7448  |
> | 4       | 50         | 0.7410   | 0.0015  | 0.7399  | 0.7453  |
> | 4       | 100        | 0.7412   | 0.0011  | 0.7404  | 0.7455  |
> |--------------------------------------
> | 8       | 5          | 0.7447   | 0.0014  | 0.7437  | 0.7452  |
> | 8       | 25         | 0.7452   | 0.0007  | 0.7450  | 0.7454  |
> | 8       | 50         | 0.7455   | 0.0002  | 0.7452  | 0.7458  |
> | 8       | 100        | 0.7456   | 0.0001  | 0.7454  | 0.7458  |
>
>
> \
> For our test results, we chose $B=32$ and 100 DDIM steps, as we did not observe further improvements in the validation results for higher values.
>
> **Supplementary material:** Kindly note that there is an Appendix with four pages of supplementary material in the main submssion, immediatly after the bibliography (not as a separate file).
>
> **Sensitivity to noise scale $F$ and classification-loss weight $\lambda$:**
> Indeed the model's performance is sensitive those parameters and also to the embedding dimension, as illustrated in the Appendix. However, we used the same hyperparameters across all of our experiments, indicating that while some tuning is helpful, it need not be dataset-specific.
>
>
>
> **Scalability to larger numbers of clusters:** Our method has no inherent algorithmic limitation on the number of clusters; the reason we restricted our experiments to up to $K = 200$ clusters was due to memory demands. Since our classification loss assumes a balanced assignment over categories, the minibatch size should grow with the number $K$ of categories for good performance. Such large big minibatch size, along with the $B$ augmented views for the diffusion student, results in high memory demand for large $K$. Nonetheless, we tested ViT-S/16 on full ImageNet ($K= 1000$) and achieved 55.95% accuracy, which is close to TEMI's accuracy [1].
>
>
> **Computation time:** The computation time increases linearly (not exponentially) with the number $B$ of diffusion samples.
> The latter can be done in parallel, unlike DDIM steps which are sequential. Thus, adding more samples $B$ does not pose a severe computational burden in practice.
>
> [1] Adaloglou et. al., Exploring the limits of deep image clustering using pretrained models, BMVC 2023.

---

### Official Review · Reviewer_o743 · 2025-03-14

**Overall Recommendation:** 1

**Summary:**

- This paper presents CLUDI, a framework that combines pre-trained Vision Transformer (ViT) features with diffusion models for clustering tasks.
   - While leveraging ViT for feature extraction and using diffusion models to enhance performance might offer some improvements, the significance of this approach could be questioned,
   - as the core feature extraction is already handled by ViT,
   - and the role of diffusion models in further boosting clustering performance remains to be fully justified.

**Claims And Evidence:**

- Most of the claims are supported clearly.

**Essential References Not Discussed:**

NAN

**Experimental Designs Or Analyses:**

- All of the experiments were well done.

**Methods And Evaluation Criteria:**

- All of the experiments were well done.

**Other Comments Or Suggestions:**

- Figures 1 and 4 do not effectively convey the motivation; they need to be improved to better showcase the core idea.

**Other Strengths And Weaknesses:**

- Strengths
   - Utilizes diffusion models for deep clustering.
- Weaknesses
  - The experimental results outperform the self-classifier, which I believe makes the comparison unfair. Alternatively, if the performance of the self-classifier is largely due to the ViT features, then this paper only presents some incremental experiments.
   - Figures 1 and 4 do not effectively convey the motivation; they need to be improved to better showcase the core idea.

**Questions For Authors:**

- Diffusion models are excellent generative algorithms, typically used for generative tasks. While the creative combination of A+B in discriminator tasks is interesting, shouldn't generative models primarily focus on generation tasks?
- The core contribution to the clustering performance seems to stem from the ViT feature rather than the diffusion model. Can you tell me the   contribution of diffusion model?

**Relation To Broader Scientific Literature:**

- DDPM (Denoising Diffusion Probabilistic Models) / VPSDE
- Contrastive learning without negative pairs
- Consistency regularization

**Theoretical Claims:**

- Most of the techniques are existing, so there is no need to further check their theoretical claims.

---

> ### Author Rebuttal · Authors · 2025-03-30
>
> We thank the reviewer for the comments and questions.
>
> **Non-incremental nature of the results:** The experimental results show that ViT features cannot by themselves explain the success of CLUDI, our model. CLUDI's advantage is evident in its superior test metrics across all models and datasets (including ImageNet subsets) we explored. Note in particular the absolute increase in accuracy of around 10% (Caltech 101 and Flower datasets) or 5.9% (Pets dataset) compared to the best of several alternative models trained on the same features, which can hardly be described as incremental.
>
> **On fairness of comparisons:** Since the core of CLUDI is a novel diffusion-based classification head (given pre-learned features), it seems that adopting the same pre-learned features across different learned classification heads is only fair for comparisons.
>
> **A novel use for diffusion models:** Although diffusion models were originally developed to generate realistic data for different modalities (images, audio, video, etc), the innovation of CLUDI resides precisely in showing that diffusion models excel in the novel task of learning to generate cluster embeddings.
>
> **Figures:** We have made special efforts to make all the figures clear and informative. Of course, we will gladly incorporate any suggestion.
>
> **The contribution of the diffusion model:** Unlike all previous self-supervised models for classification, CLUDI relies on a probabilistic formulation based on the marginalization of latent diffusion variables. This probabilistic component is the main contribution of the diffusion model and arguably the reason for CLUDI's robustness (see Figure 2) and state-of-the-art performance (see Tables 1 and 2).

---

### Decision · Program_Chairs · 2025-05-01

**Decision:**

Accept (poster)

**Comment:**

This paper proposes CLUDI, a novel self-supervised image clustering framework that integrates diffusion models with Vision Transformer (ViT) features in a teacher-student setup. The method aims to generate diverse and robust cluster assignments, and is validated on several standard benchmarks, achieving competitive performance.

Reviewers appreciated the novelty of applying diffusion models to clustering tasks, noting that this could inspire future directions in unsupervised representation learning. The paper is clearly written, well-organized, and presents strong empirical results on CIFAR-10, STL-10, and subsets of ImageNet.

However, concerns were raised about the extent to which the diffusion model contributes beyond the already powerful ViT features. In particular, the method's sensitivity to hyperparameters and the unclear impact of diffusion step count at inference time may limit its practicality and generalizability. The dependence on careful tuning across datasets could make the method harder to deploy in real-world unsupervised settings.

Overall, the paper offers an interesting and novel perspective, but further analysis is needed to better isolate and justify the benefits of the diffusion component and to improve robustness across datasets.